# TriangleMix: Accelerating Prefilling via Decoding-time Contribution Sparsity

## Abstract

Large Language Models (LLMs) incur quadratic attention complexity with input length, creating a major time bottleneck in the prefilling stage. Existing acceleration methods largely exploit *attention score sparsity* by estimating blocks with high attention scores and applying dynamic sparse attention. In this work, we identify another untapped form of sparsity in the prefilling stage, namely *decoding-time contribution sparsity*, where many attention blocks exhibit nontrivial attention scores during prefilling yet contribute negligibly to subsequent decoding, as indicated by gradient-based analysis. Building on this observation, we propose TriangleMix, a training-free static attention pattern that uses dense attention in a subset of layers and switches to Triangle attention in the others. Extensive experiments show that TriangleMix preserves nearly lossless performance relative to dense attention while substantially reducing attention overhead in Triangle layers. For 128K inputs, Triangle attention achieves a $15.3\times$ speedup in attention computation, significantly exceeding the acceleration of typical dynamic sparse methods ($1.9\times$–$3.4\times$). Furthermore, TriangleMix can be seamlessly combined with dynamic sparsity approaches, delivering an additional 6%–19% reduction in TTFT over using dynamic sparsity alone.

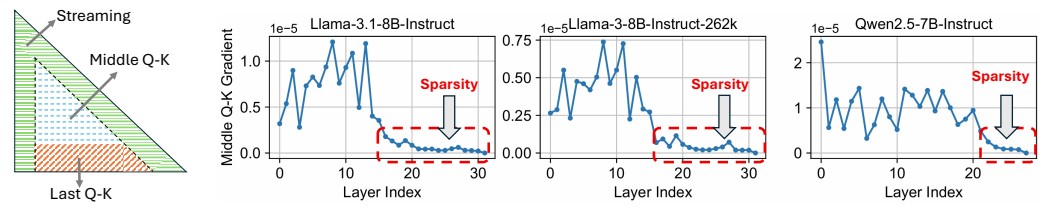

(a) Attention sections.

(b) Gradient of Middle Q-K with respect to the first generated token

Figure 1: The average gradient $\mathrm{Grad}(M, l)$ of the Middle Q-K sections, measured on three models, shows a significant sparsity in deeper layers. This suggests that *the Middle Q-K components in deeper layers contribute minimally to decoding* and might potentially be skipped to improve efficiency.

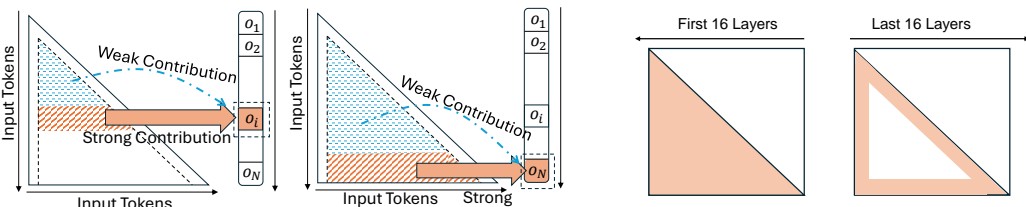

(a) Measure contribution based on different tokens

(b) TriangleMix on Llama-3.1-8B-Instruct

Figure 2: **Left:** Attention computation in certain layers exhibits contribution locality. **Right:** The proposed TriangleMix pattern for Llama-3.1-8B-Instruct.

# 1 INTRODUCTION

Large Language Models (LLMs) are capable of processing input sequences of varying lengths (Grattafiori et al., 2024; Achiam et al., 2023; Yang et al., 2024), a crucial ability that supports diverse downstream tasks, such as question answering (Bai et al., 2023), long-form document understanding (Zhao et al., 2024), and code generation (Jiang et al., 2024b). However, due to the quadratic complexity of attention mechanisms, the attention computation time significantly increases as the input context length grows. As prior research has demonstrated, attention computation has become a critical bottleneck in the prefilling stage of LLMs (Jiang et al., 2024a; Lai et al., 2025).

To mitigate this bottleneck, several dynamic sparse attention techniques have been proposed, including MInference (Jiang et al., 2024a), FlexPrefill (Lai et al., 2025), and XAttention (Xu et al., 2025). These methods exploit *attention score sparsity* to accelerate computation: the majority of entries in attention matrix have negligible values and can be safely skipped, while blocks with significant scores need to be computed. Dynamic sparsity methods first estimate which regions are likely to contain high attention scores, and then selectively perform computation on these regions to approximate full attention (Jiang et al., 2024a; Lai et al., 2025; Xu et al., 2025).

Dynamic sparsity attention is built on a simple intuition: low-score blocks can be skipped, while blocks with non-trivial scores must be computed. This raises a deeper question: **Must we compute all non-trivial blocks during prefilling to preserve decoding accuracy?** A natural concern is that skipping them might distort the attention distribution and degrade performance. However, our analysis reveals that the situation is subtler. We uncover a new form of sparsity in the prefilling stage, which we call *decoding-time contribution sparsity*. Unlike score sparsity, it reflects that certain blocks, though holding non-trivial scores, have little impact on the actual decoding process.

To rigorously evaluate the necessity of each block, we analyze how its removal affects the generation of the first token following the prompt. Gradient analysis offers a direct measure of this sensitivity. As shown in Figure 1a, we divide causal attention into three regions: Last Q–K, Middle Q–K, and the Streaming region. Our results indicate that the Streaming and Last Q–K regions are critical to decoding, while the Middle Q–K exhibits significant layer-wise sparsity pattern (Figure 1b). Notably, although the attention score magnitude of Middle Q–K regions are often comparable to Last Q–K, its gradients reveal substantially lower impact in certain layers.

This sparsity arises from the phenomenon that certain parts of attention scores' contribution has *locality of influence* with respect to the decoding token position. As shown in Figure 2a, many attention scores only affect predictions within a limited temporal window after they occur. The Middle Q-K region in some layers has considerable influence on an intermediate token $o_i$, but contribute very little to the tokens generated after the entire prompt $o_N, o_{N+1}, ....$ This distinction allows us to safely skip Middle Q-K region's attention computation of certain layers during the prefilling phase.

Motivated by this insight, we introduce **TriangleMix**, a simple yet effective static attention pattern for accelerating prefilling. As depicted in Figure 2b, TriangleMix combines dense attention in some layers with a triangle-shaped sparse attention pattern in others. This design brings four key benefits:

- **Training-free:** It can be applied directly to state-of-the-art pretrained LLMs without fine-tuning.
- **Nearly lossless accuracy:** Despite its simplicity, TriangleMix preserves model performance, reaching accuracy comparable to sophisticated dynamic sparsity methods.
- **Efficiency:** Its static triangle pattern removes the need for block index estimation, reducing complexity from $O(N^2)$ to $O(N)$ and enabling significantly simpler and faster attention kernel than dynamic sparsity attention.
- **Complementary to dynamic attention:** Replacing dynamic sparsity with Triangle attention in selected layers delivers extra acceleration while maintaining performance.

We conduct extensive experiments on three long-context LLMs, including Llama-3.1-8B-Instruct (Grattafiori et al., 2024), Llama-3-8B-Instruct-262K (GradientAI, 2024), and Qwen2.5-7B-Instruct (Yang et al., 2024), using all tasks from the RULER and LongBench benchmarks (Hsieh et al., 2024; Bai et al., 2023). Our results show that TriangleMix preserves the accuracy of full attention while substantially improving efficiency. Specifically, with 128K-token inputs, Triangle attention achieves a $15.3\times$ speedup in attention computation, far exceeding typical dynamic sparse methods ($1.9\times$–$3.4\times$). For Llama-3.1-8B-Instruct, TriangleMix reduces overall TTFT by 12%–32% across context lengths

from 32K to 128K. In addition, TriangleMix integrates seamlessly with dynamic sparsity approaches, yielding a further 6% to 19% decrease in TTFT compared to dynamic sparsity alone.

## 2 METHODOLOGY

### 2.1 PROBING ATTENTION BLOCK CONTRIBUTION

The prefilling stage of Transformer attention can be formulated as:

$$\boldsymbol{A} = \text{Softmax}(\frac{1}{\sqrt{d}}\boldsymbol{Q}\boldsymbol{K}^T - c(1 - \boldsymbol{M}))$$

where $\boldsymbol{Q}, \boldsymbol{K}, \boldsymbol{V}$ are matrices of shape $(N, d)$, and $\boldsymbol{M}$ is a causal mask matrix of shape $(N, N)$, with entries $\boldsymbol{M}_{i,j} \in \{0, 1\}$. Here, $N$ represents the number of input tokens, and $c$ is a large positive constant to ensure attention scores masked by $\boldsymbol{M}_{i,j} = 0$ become effectively zero after the softmax operation. To accelerate computing, sparse attention (Jiang et al., 2024a; Lai et al., 2025; Xu et al., 2025) aims to find a sparse mask matrix $\boldsymbol{M}'$ to compute the attention output:

$$\boldsymbol{A}' = \text{Softmax}(\frac{1}{\sqrt{d}}\boldsymbol{Q}\boldsymbol{K}^T - c(1 - \boldsymbol{M}'))$$

The mask matrix $\boldsymbol{M}'$ can be either static or dynamic. StreamingLLM (Xiao et al., 2023) and LM-infinite (Han et al., 2024) employ similar static streaming mask, which restricts attention to a few sink tokens and a sliding window of nearby tokens. Such pattern reduces the computation complexity of attention to $O(N)$ but significantly harms the performance of long-context LLMs(Li et al., 2024a).

In contrast, dynamic masks enable dynamic sparsity attention. During inference, block indices in $\boldsymbol{M}'$ that likely yield non-trivial scores are identified from the input, and FlashAttention (Dao et al., 2022) is applied only to these blocks. Methods such as MInference (Jiang et al., 2024a), FlexPrefill (Lai et al., 2025), and XAttention (Xu et al., 2025) all adopt this strategy.

A central challenge for these dynamic sparsity attention lies in efficiently and accurately estimating the block indices with meaningful attention scores. MInference categorizes attention heads into three types and applies different estimation strategies, either by observing recent query tokens or by pooling vectors from $\boldsymbol{Q}$ and $\boldsymbol{K}$. FlexPrefill adopts a similar strategy but determines the head type dynamically during inference. XAttention, on the other hand, uses the sum of anti-diagonal scores to identify blocks with large attention. Despite these differences, the common objective remains the same: reliably locating blocks with meaningful attention scores. In this paper, we move beyond block index estimation and pose a more fundamental question: **Must we compute all non-trivial blocks during prefilling to preserve decoding accuracy?**

To answer this, we quantitatively probe each block's contribution relative to the generated tokens in decoding, which we call its ***decoding-time contribution***. Our goal is to prioritize computation for blocks with higher decoding-time contribution. Since gradients naturally capture such sensitivity, we propose a gradient-based method to perform this analysis.

Given an input $\boldsymbol{X}_{\text{input}}$ containing $N$ tokens fed into a language model, we define a probing variable $\boldsymbol{\theta} \in \mathbb{R}^{L \times N \times N}$, where $\boldsymbol{\theta}_{\ell,i,j} = 1$ for all layers $\ell$ and token indices $i, j$. Here, $L$ represents the total number of layers in the model. For simplicity of notation, we omit the attention head dimension here, but our proposed method generalizes naturally to both Multi-head Attention (Vaswani et al., 2017) and Grouped Query Attention (Ainslie et al., 2023).

In layer $l$, the attention scores is calculated as:

$$\boldsymbol{A}_{\boldsymbol{\theta}} = \boldsymbol{\theta}_l \odot \text{Softmax}\left(\frac{1}{\sqrt{d}}\boldsymbol{Q}\boldsymbol{K}^T - c(1 - \boldsymbol{M})\right)$$

where $\odot$ denotes element-wise multiplication. Since all elements of $\boldsymbol{\theta}$ are initially set to 1, this operation does not alter the attention scores.

The model then outputs a logit prediction $\boldsymbol{Y_\theta}$ for the next token after the $N$th token, represented as a vector $\boldsymbol{Y_\theta} = [y_1, y_2, \ldots, y_{\text{vocab}}]$. Note that it is only the output of the last query. Our focus

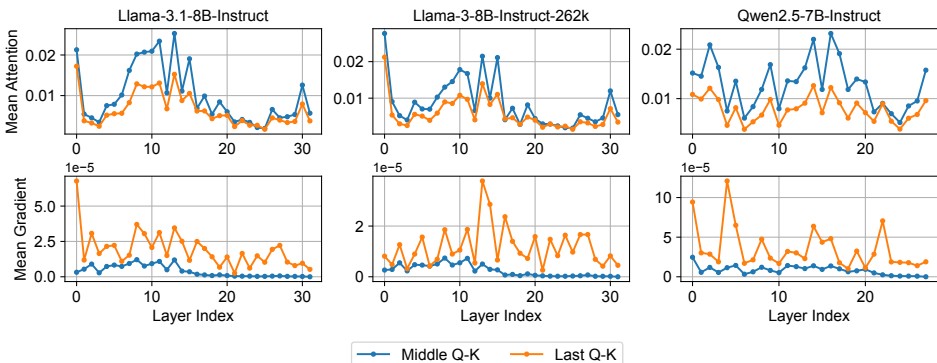

Figure 3: First row: Average attention score for the Middle and Last Q-K sections; Second row: average gradient $\mathrm{Grad}(\boldsymbol{M}, l)$ for the Middle and Last Q-K sections.

is on a specific logit $y_{\mathrm{gt}}$, associated with the correct label or ground truth token. For instance, the correct answer in a multiple-choice scenario or the initial token of the target sequence in a needle-in-a-haystack task. We are particularly interested in the partial derivative $\frac{\partial y_{\mathrm{gt}}}{\partial \boldsymbol{\theta}_{l,i,j}}$, which measures the sensitivity of the model's output to changes in attention scores. We quantify the importance of a specific attention section $\boldsymbol{M}'$ by computing the mean of the derivative values within that section. Formally, we define the gradient-based importance as:

$$\mathrm{Grad}(\hat{\boldsymbol{M}}, l) = \frac{\sum_{\hat{\boldsymbol{M}}_{i,j}=1} \frac{\partial y_{\mathrm{gt}}}{\partial \boldsymbol{\theta}_{l,i,j}}}{\sum_{\hat{\boldsymbol{M}}_{i,j}=1} \hat{\boldsymbol{M}}_{i,j}}$$

Here, $\hat{\boldsymbol{M}} \in \{0,1\}^{N \times N}$ is a binary mask specifying the attention section of interest, with $\hat{\boldsymbol{M}}_{i,j} = 1$ indicating that the pair $(i,j)$ belongs to the region. The quantity $\mathrm{Grad}(\hat{\boldsymbol{M}}, l)$ thus represents the *average sensitivity of $y_{\mathrm{gt}}$ to perturbations within region $\hat{\boldsymbol{M}}$ at layer $l$*. This formulation is general and can be applied to quantify the contribution of any region in the attention map.

## 2.2 DECODING-TIME SPARSITY IN MIDDLE Q-K SECTIONS

Under the framework proposed in Section 2.1, we conduct an initial gradient-based analysis of three predefined attention sections. As illustrated in Figure 1a, we partition attention into the Streaming section, the Middle Q–K section, and the Last Q–K section. We evaluate these regions using a key–value retrieval task in which the language model must output the corresponding value beginning with its first generated token; we set $y_{\mathrm{gt}}$ to the first token of the correct value. See Appendix A.1 for the exact region definitions and the full prompt.

First, we observe that both the attention scores and the gradients are high in the Streaming section, consistent with prior works (Xiao et al., 2023; Han et al., 2024). For the Middle Q–K and Last Q–K sections, we summarize our findings as follows:

**Slightly higher attention scores in Middle Q–K.** As shown in the top row of Figure 3, the average attention scores in the Middle Q–K and Last Q–K sections are similar, with the Middle Q–K section being slightly higher. This indicates that both regions contain non-trivial attention blocks.

**Lower gradients in Middle Q–K.** Despite the slightly higher attention scores, the Middle Q–K section exhibits substantially smaller gradients than the Last Q–K section (The bottom row of Figure 3). This suggests that, for predicting the first token, computing blocks in the Last Q–K region is more critical.

**Layer-wise sparsity.** The gradients within the Middle Q–K section display clear layer-wise sparsity. For example, in Llama-3.1-8B-Instruct and Llama-3-8B-262K (Figure 1b), the gradients of the last 16 layers are near zero, implying that block computations in this region could be skipped with minimal impact on first-token prediction. We refer to this phenomenon as *decoding-time contribution sparsity*, since it is defined directly by contributions to decoded tokens. Moreover, the observation extends

beyond the first token: for later generated tokens, the current Middle Q–K region becomes a subset of the future Middle Q–K region.

## 2.3 PERPLEXITY ANALYSIS

| Method | Llama-3.1-8B-Instruct | | Llama-3-Instruct-262k | | Qwen2.5-7B-Instruct | |
|---|---|---|---|---|---|---|
| | 1024–1152 | 1920–2048 | 1024–1152 | 1920–2048 | 1024–1152 | 1920–2048 |
| Dense | 8.31 | 7.81 | 8.05 | 7.62 | 8.05 | 7.62 |
| Skip Q (512–1024)-K 50% layers | 8.58 (+0.27) | 7.82 (+0.01) | 8.33 (+0.28) | 7.63 (+0.01) | 8.16 (+0.11) | 7.64 (+0.02) |
| Skip Q (512–1024)-K all layers | 8.76 (+0.45) | 7.90 (+0.09) | 8.54 (+0.49) | 7.72 (+0.10) | 8.54 (+0.49) | 7.72 (+0.10) |

Table 1: Perplexity (PPL) comparison across different models and settings.

To further validate our claim, we study the impact of removing attention computation in the middle region. We randomly sample 1,024 Wikipedia articles and take the first 2,048 tokens from each. Three settings are compared: (1) full attention, (2) skipping attention with Q indices from 512 to 1,024 across all layers, and (3) skipping the same region in only half of the layers. In both (2) and (3), the Streaming section and the attention with other Q indices are still retained. For the third setting, we select the half of layers with the lowest $\text{Grad}(\hat{M}, l)$, where $\hat{M}$ denotes the skipped attention region and the gradient measures its contribution to predicting the 2049th token. For each model, the skipped layers remain fixed across all 1,024 samples, although different models may choose different sets of layers.

Table 1 reports perplexity changes for token ranges 1024–1152 and 1920–2048. At position 1024, the skipped computation corresponds to the Last Q–K section (since Q indices 512–1024 are omitted). At positions 1920–2048, the skipped region aligns with the Middle Q–K section. We find that skipping either all or half of the layers significantly increases perplexity in the 1024–1152 range, confirming our observations in Section 2.2 that the Last Q–K section is critical. In contrast, at 1920–2048, skipping half of the layers causes only a minor increase in perplexity (0.01–0.02), though skipping all Middle Q–K computations still raises perplexity noticeably.

We visualize this phenomenon in Figure 2a. We refer to it as the ***locality of influence***, which describes the finite-horizon effect of certain attention scores in transformer models. Specifically, even if some attention blocks have non-trivial scores, their influence on the output distribution disappears once the decoding position $n$ exceeds $t + \Delta t$, where $t$ is the query index of the block and $\Delta t$ is a small constant (e.g., $\Delta t = 128$). This indicates that many attention scores only affect predictions within a limited temporal window after they occur. As a result, they influence nearby predictions but do not affect distant tokens, particularly those beyond the prompt. This distinction allows us to safely skip computing certain attention weights during the prefilling phase without relying on the magnitude of the attention weights themselves.

## 2.4 TRIANGLEMIX

Based on these observations, we introduce **TriangleMix**, an effective and efficient static attention pattern for long-context prefilling. The key idea is that the Middle Q–K section in some layers contributes little to decoding, and can therefore be safely skipped. In these layers, attention reduces to the *Triangle* pattern shown in Figure 2b. This modification lowers the complexity of attention in those layers from $O(N^2)$ to $O(N)$. Unlike dynamic sparsity, Triangle attention is static: there is no need to predict block indices or design specialized sparse kernels. As a result, the kernel is much simpler and faster to implement. Overall, TriangleMix combines dense attention in some layers with Triangle attention in others.

For a given model, the application of TriangleMix is as follows. We first conduct the gradient-based analysis introduced in Sections 2.1 and 2.2. For each layer $l_i$, we compute the decoding-time contribution of its Middle Q–K section, denoted as $\text{Grad}(M^{\text{middle}}, l_i)$ for $i = 1, 2, \ldots, N_{\text{layer}}$. Based on these values, we identify the $L_{\text{tri}}$ layers with the lowest contributions, where $L_{\text{tri}}$ is a hyperparameter controlling how many layers are converted into Triangle attention. These selected layers adopt the Triangle pattern at inference time, thereby reducing computation. The remaining layers retain their original dense attention. Alternatively, they can also be replaced with dynamic sparse attention methods to achieve further acceleration.

## 3 EVALUATIONS

### 3.1 SETTINGS

**LLMs and Benchmarks.** We evaluate our method on three recent long-context LLMs: Llama-3.1-8B-Instruct (Grattafiori et al., 2024), Llama-3-8B-Instruct-262K (GradientAI, 2024), and Qwen2.5-7B-Instruct (Yang et al., 2024). We use two challenging benchmarks: LongBench (Bai et al., 2023) and RULER (Hsieh et al., 2024). On RULER, we test input lengths of 4K, 8K, 16K, 32K, 64K, and 128K. On LongBench, we evaluate all English tasks.

**Hyperparameters.** Different models exhibit varying levels of decoding-time contribution sparsity. We set $L_{\text{tri}} = 16$ for Llama-3.1-8B-Instruct and Llama-3-8B-Instruct-262K, and $L_{\text{tri}} = 8$ for Qwen2.5-7B-Instruct, which preserves nearly lossless accuracy. An analysis of different $L_{\text{tri}}$ settings is provided in Section 3.3. For all experiments, we fix the sink token size to 8 and the local window size to 512.

**Static Sparsity Baselines.** We compare our approach with four static sparsity baselines: (1) **Streaming pattern** (Xiao et al., 2023; Han et al., 2024), applied to all layers during prefilling; (2) **Triangle attention**, also applied to all layers; (3) **StreamingMix**, where Triangle attention in TriangleMix is replaced by the Streaming pattern; (4) **DuoAttention** (Xiao et al., 2024), which learns a separate sparsity pattern for each attention head. For DuoAttention, we use the official pattern for Llama-3.1-8B-Instruct and train a pattern for Llama-3-8B-Instruct-262K using the authors' script. We set the sparsity ratio to be 50%. However, training on Qwen2.5-7B-Instruct did not converge, so results are omitted. For all static sparsity baselines, we set the sink token number and local window to 8 and 512, consistent with our method.

| Methods | 4K | 8K | 16K | 32K | 64K | 128K | Avg. |
|---|---|---|---|---|---|---|---|
| *Llama-3.1* | 96.6 | 95.3 | 94.8 | 91.3 | 86.3 | 78.1 | 90.4 |
| Streaming | 64.1 | 55.4 | 40.5 | 28.9 | 26.7 | 3.3 | 36.5 |
| Triangle | 88.8 | 88.3 | 82.8 | 72.6 | 65.0 | 39.0 | 72.7 |
| StreamingMix | 94.3 | 91.8 | 90.2 | 86.2 | 79.2 | 71.9 | 85.6 |
| DuoAttention | 95.7 | 93.1 | 88.4 | 84.3 | 82.5 | 64.0 | 84.7 |
| TriangleMix | **96.3** | **95.1** | **94.7** | **91.3** | **86.3** | **77.5** | **90.2** |
| *Llama-3-262k* | 93.4 | 90.3 | 88.8 | 85.1 | 82.2 | 79.4 | 86.5 |
| Streaming | 49.4 | 38.7 | 33.4 | 30.2 | 26.5 | 21.7 | 33.3 |
| Triangle | 88.1 | 84.7 | 80.6 | 71.0 | 65.0 | 55.4 | 74.1 |
| StreamingMix | 87.3 | 83.2 | 80.8 | 79.9 | 77.6 | 70.8 | 79.9 |
| DuoAttention | 92.8 | **91.9** | **89.0** | **85.0** | 81.1 | 76.1 | 86.0 |
| TriangleMix | **93.5** | 91.0 | 88.1 | **85.0** | 82.4 | 79.6 | 86.6 |
| *Qwen2.5* | 95.8 | 93.6 | 92.6 | 84.5 | 81.9 | 67.4 | 86.0 |
| Streaming | 55.6 | 47.8 | 35.9 | 30.6 | 28.1 | 21.0 | 36.5 |
| Triangle | 88.4 | 81.9 | 74.0 | 58.3 | 51.0 | 14.6 | 61.4 |
| StreamingMix | 93.3 | 90.4 | 89.4 | 81.7 | 76.4 | 63.6 | 82.5 |
| TriangleMix | **95.5** | **93.8** | **92.1** | **84.6** | **80.2** | **67.7** | **85.6** |

Table 2: Comparison with static sparsity methods on RULER. Llama-3.1, Llama-3-262K, Qwen2.5 are abbreviations for Llama-3.1-8B-Instruct, Llama-3-8B-262K, and Qwen2.5-8B-Instruct. The same applies to Table 3.

| Methods | 4K | 8K | 16K | 32K | 64K | 128K | Avg. |
|---|---|---|---|---|---|---|---|
| *Llama-3.1* | 96.6 | 95.3 | 94.8 | 91.3 | 86.3 | 78.1 | 90.4 |
| TriangleMix | 96.3 | 95.1 | 94.7 | 91.3 | 86.3 | 77.5 | 90.2 |
| MInference | 96.3 | 95.1 | 95.0 | 90.5 | 86.8 | 75.0 | 89.8 |
| Ours + MInfer | 96.2 | 95.0 | 94.5 | 90.8 | 87.2 | 75.8 | 89.9 |
| FlexPrefill | 95.0 | 94.7 | 94.5 | 92.8 | 86.5 | 75.9 | 89.9 |
| Ours + FP | 95.2 | 95.0 | 94.7 | 92.8 | 86.8 | 76.2 | 90.1 |
| XAttn | 96.5 | 94.9 | 94.5 | 91.8 | 85.7 | 73.8 | 89.5 |
| Ours + XAttn | 96.2 | 95.1 | 94.9 | 91.9 | 85.7 | 72.9 | 89.4 |
| *Llama-3-262k* | 93.4 | 90.3 | 88.8 | 85.1 | 82.2 | 79.4 | 86.5 |
| TriangleMix | 93.5 | 91.0 | 88.1 | 85.0 | 82.4 | 79.6 | 86.6 |
| MInference | 93.4 | 90.7 | 88.8 | 85.1 | 82.6 | 80.1 | 86.8 |
| Ours + MInfer | 93.0 | 90.4 | 88.6 | 84.4 | 82.7 | 80.1 | 86.5 |
| FlexPrefill | 90.3 | 87.7 | 88.0 | 83.8 | 80.2 | 75.7 | 84.3 |
| Ours + FP | 90.2 | 87.3 | 87.3 | 83.8 | 80.0 | 76.6 | 84.2 |
| XAttn | 93.6 | 91.1 | 87.7 | 84.8 | 82.6 | 78.9 | 86.5 |
| Ours + XAttn | 93.3 | 91.2 | 87.9 | 84.3 | 82.5 | 79.3 | 86.4 |
| *Qwen2.5* | 95.8 | 93.6 | 92.6 | 84.5 | 81.9 | 67.4 | 86.0 |
| TriangleMix | 95.5 | 93.8 | 92.1 | 84.6 | 80.2 | 67.7 | 85.6 |
| MInference | 95.6 | 93.8 | 92.8 | 84.9 | 79.1 | 68.0 | 85.7 |
| Ours + MInfer | 95.7 | 93.5 | 92.4 | 84.8 | 76.9 | 63.5 | 84.5 |
| FlexPrefill | 91.1 | 89.7 | 88.5 | 75.4 | 72.5 | 51.2 | 78.1 |
| Ours + FP | 92.2 | 91.1 | 89.5 | 77.0 | 73.3 | 52.7 | 79.3 |
| XAttn | 95.4 | 93.7 | 92.0 | 82.9 | 77.8 | 66.2 | 84.7 |
| Ours + XAttn | 95.3 | 93.3 | 91.5 | 82.2 | 77.4 | 64.9 | 84.1 |

Table 3: Comparison with dynamic sparsity methods on RULER.

**Dynamic Sparsity Baselines.** We further compare with three dynamic sparsity methods: **MInference** (Jiang et al., 2024a), **FlexPrefill** (Lai et al., 2025), and **XAttention** (Xu et al., 2025). Our method can also be combined with these approaches: Triangle attention is retained, while layers that require full attention are replaced with dynamic sparse attention. This design further reduces runtime overhead compared with dynamic sparsity alone, since Triangle attention skips more attention computation and is much simpler. We denote the combined versions as "Ours + MInference," "Ours + FlexPrefill," and "Ours + XAttention." For dynamic methods, we also set their hyperparameters to maintain near-lossless accuracy: $\gamma = 0.95$ for FlexPrefill, and $\tau = 0.95$ with $\text{stride} = 8$ for XAttention.

## 3.2 EFFECTIVENESS OF TRIANGLEMIX

**Comparison with Static Sparsity.** Table 2 and Table 5 present the evaluation results of TriangleMix and various static sparsity baselines. TriangleMix consistently outperforms all static baselines and remains nearly lossless performance across all input lengths. In contrast, other static methods usually suffer noticeable performance degradation. DuoAttention performs comparably to TriangleMix on LongBench using Llama-3.1-8B-Instruct. However, its performance is worse than TriangleMix on RULER, especially deteriorates at longer lengths (e.g., 64K and 128K). Furthermore, DuoAttention requires a separate training phase to learn head-wise sparsity, which introduces additional computational overhead and may fail to converge on models such as Qwen-2.5-7B-Instruct.

On the LongBench benchmark, we observe that the full Triangle pattern performs poorly on PassageRetrieval tasks, while the StreamingMix pattern underperforms on the in-context learning task (TREC). These findings suggest that attention over the Middle Q-K section in certain layers (especially shallow layers) is still crucial for retrieval tasks, while attention over the Last Q-K section in certain layers (especially deep layers) plays a key role in supporting in-context learning.

| Methods | Average | 2Wiki | GovRep | Hotpot | LCC | MNews | MF-en | PsgCnt | PsgRtr | Qasper | Rbench | Samsum | TREC | TrvQA |
|---|---|---|---|---|---|---|---|---|---|---|---|---|---|---|
| *Llama-3.1-8B-Instruct* | 54.8 | 48.1 | 34.4 | 61.6 | 66.6 | 25.6 | 56.9 | 16.8 | 99.7 | 44.9 | 52.7 | 42.6 | 71.0 | 91.6 |
| TriangleMix | 54.5 | 47.8 | 34.3 | 61.5 | 67.0 | 25.8 | 56.5 | 16.8 | 98.3 | 44.8 | 51.9 | 42.2 | 69.7 | 92.2 |
| MInference | 54.9 | 48.1 | 34.5 | 61.4 | 67.0 | 25.9 | 57.0 | 17.5 | 99.7 | 44.7 | 52.5 | 42.6 | 71.3 | 91.9 |
| Ours + MInference | 54.5 | 48.3 | 34.2 | 62.0 | 66.7 | 25.6 | 56.6 | 17.4 | 98.3 | 44.8 | 51.6 | 42.1 | 69.3 | 92.1 |
| FlexPrefill | 48.4 | 39.4 | 33.7 | 58.3 | 67.2 | 25.6 | 55.6 | 4.0 | 47.7 | 42.5 | 53.8 | 41.9 | 69.7 | 90.2 |
| Ours + FlexPrefill | 52.2 | 39.4 | 33.5 | 58.5 | 66.9 | 25.7 | 54.5 | 3.3 | 97.0 | 43.1 | 53.8 | 42.3 | 69.3 | 90.8 |
| XAttention | 53.5 | 45.4 | 34.5 | 61.0 | 66.9 | 25.7 | 55.9 | 18.1 | 86.0 | 42.9 | 55.5 | 42.9 | 71.0 | 90.2 |
| Ours + XAttention | 54.3 | 45.0 | 34.6 | 61.0 | 66.8 | 25.6 | 57.1 | 19.0 | 98.7 | 42.4 | 54.4 | 42.0 | 69.3 | 90.4 |
| *Llama-3-8B-Instruct-262k* | 44.2 | 21.0 | 34.3 | 26.4 | 46.1 | 26.3 | 50.2 | 4.7 | 95.0 | 30.5 | 43.3 | 41.1 | 68.3 | 86.8 |
| TriangleMix | 43.4 | 20.7 | 34.5 | 25.7 | 47.3 | 26.1 | 51.1 | 4.7 | 85.3 | 30.7 | 43.6 | 40.9 | 66.7 | 87.0 |
| MInference | 44.1 | 20.5 | 34.3 | 25.8 | 46.2 | 26.3 | 50.2 | 4.3 | 95.0 | 30.7 | 44.7 | 40.5 | 68.3 | 86.8 |
| Ours + MInference | 43.4 | 20.6 | 34.4 | 26.4 | 47.2 | 26.2 | 51.1 | 4.3 | 85.3 | 30.3 | 43.6 | 40.9 | 66.7 | 86.9 |
| FlexPrefill | 35.0 | 17.7 | 33.0 | 25.1 | 36.8 | 26.1 | 49.1 | 3.3 | 13.7 | 28.3 | 36.2 | 39.1 | 65.0 | 81.7 |
| Ours + FlexPrefill | 36.0 | 18.3 | 33.2 | 25.1 | 36.9 | 26.2 | 49.6 | 6.7 | 24.0 | 28.3 | 35.6 | 38.9 | 64.3 | 80.4 |
| XAttention | 41.5 | 20.7 | 34.3 | 26.0 | 37.9 | 26.1 | 52.3 | 4.7 | 71.3 | 30.8 | 39.4 | 40.0 | 68.3 | 87.2 |
| Ours + XAttention | 42.1 | 21.1 | 34.3 | 26.1 | 37.7 | 26.2 | 51.7 | 5.7 | 81.3 | 30.3 | 38.3 | 39.9 | 67.3 | 87.0 |
| *Qwen2.5-7B-Instruct* | 46.8 | 24.5 | 30.7 | 32.2 | 62.2 | 22.3 | 41.3 | 13.1 | 99.3 | 24.4 | 60.7 | 42.5 | 67.0 | 88.7 |
| TriangleMix | 46.1 | 24.7 | 30.6 | 32.3 | 62.6 | 22.3 | 40.9 | 13.1 | 89.3 | 23.9 | 59.6 | 42.4 | 69.3 | 88.5 |
| MInference | 46.8 | 24.5 | 30.7 | 32.7 | 61.6 | 22.3 | 40.8 | 12.8 | 100.0 | 24.3 | 60.2 | 42.1 | 68.0 | 89.3 |
| Ours + MInference | 46.1 | 25.7 | 30.3 | 32.2 | 61.9 | 22.3 | 40.7 | 11.4 | 89.7 | 25.8 | 59.4 | 42.5 | 68.7 | 89.0 |
| FlexPrefill | 40.8 | 20.2 | 29.7 | 33.9 | 66.8 | 21.8 | 40.4 | 4.0 | 48.7 | 17.1 | 54.8 | 40.5 | 67.3 | 85.5 |
| Ours + FlexPrefill | 41.7 | 21.0 | 29.8 | 32.8 | 64.8 | 21.9 | 39.6 | 5.3 | 56.0 | 19.8 | 55.1 | 40.9 | 69.0 | 85.8 |
| XAttention | 46.6 | 22.4 | 30.9 | 34.0 | 63.9 | 22.4 | 40.5 | 10.9 | 98.0 | 23.5 | 60.9 | 42.2 | 67.7 | 88.8 |
| Ours + XAttention | 45.8 | 23.6 | 31.1 | 33.1 | 63.8 | 21.7 | 41.3 | 9.8 | 88.3 | 24.5 | 58.2 | 42.4 | 69.0 | 88.4 |

Table 4: Comparison with dynamic sparsity methods on LongBench.

**Comparison with Dynamic Sparsity.** Tables 3 and 4 present the results of TriangleMix compared with dynamic sparsity methods, as well as their combinations. Despite its simplicity, TriangleMix maintains accuracy comparable to dynamic sparsity methods. On the RULER benchmark, it yields an average accuracy change of only $-0.4\%$ to $+0.1\%$ relative to full attention, which is on par with MInference ($-0.6\%$ to $+0.3\%$), and outperforms FlexPrefill ($-9.1\%$ to $-0.6\%$) and XAttention ($-2.2\%$ to $+0.0\%$). Importantly, although TriangleMix skips many attention blocks with non-trivial scores, the decoding accuracy remains nearly lossless. This supports our claim that such blocks contribute only marginally to decoding-time accuracy, and that decoding-time sparsity is a general property across all three tested models.

Moreover, when combined with dynamic sparsity methods such as MInference, FlexPrefill, or XAttention, TriangleMix consistently preserves their original performance. On LongBench, combining TriangleMix with FlexPrefill or XAttention even yields slight performance improvements. These combinations are meaningful: by eliminating computation in many additional attention blocks, TriangleMix reduces the computation required by dynamic sparsity methods, making prefilling even faster than using dynamic sparsity alone.

Overall, TriangleMix preserves nearly full accuracy of dense attention across tasks while significantly reducing time complexity from $O(N^2)$ to $O(N)$ in layers using Triangle attention.

| Methods | Average | 2Wiki | GovRep | Hotpot | LCC | MNews | MF-en | PsgCnt | PsgRtr | Qasper | Rbench | Samsum | TREC | TrvQA |
|---|---|---|---|---|---|---|---|---|---|---|---|---|---|---|
| *Llama-3.1-8B-Instruct* | 54.8 | 48.1 | 34.4 | 61.6 | 66.6 | 25.6 | 56.9 | 16.8 | 99.7 | 44.9 | 52.7 | 42.6 | 71.0 | 91.6 |
| Streaming | 34.7 | 17.4 | 32.2 | 19.8 | 65.6 | 24.6 | 25.6 | 5.7 | 11.8 | 23.3 | 54.1 | 40.6 | 52.7 | 77.3 |
| Triangle | 47.0 | 40.6 | 33.2 | 52.3 | 63.6 | 24.6 | 53.5 | 7.9 | 42.0 | **46.3** | 46.5 | **43.0** | 67.0 | 90.1 |
| StreamingMix | 48.9 | 36.4 | **34.3** | 39.4 | 64.9 | **25.8** | 42.3 | **18.4** | 98.3 | 42.4 | 53.6 | 42.6 | 51.0 | 86.8 |
| DuoAttention | **54.5** | 44.6 | 34.1 | 59.1 | **68.0** | 25.3 | 55.0 | 15.7 | **99.7** | 43.9 | **58.8** | 42.7 | **71.3** | 91.0 |
| TriangleMix | **54.5** | **47.8** | 34.3 | **61.5** | 67.0 | 25.8 | 56.5 | 16.8 | 98.3 | 44.8 | 51.9 | 42.2 | 69.7 | **92.2** |
| *Llama-3-8B-Instruct-262k* | 44.2 | 21.0 | 34.3 | 26.4 | 46.1 | 26.3 | 50.2 | 4.7 | 95.0 | 30.5 | 43.3 | 41.1 | 68.3 | 86.8 |
| Streaming | 30.8 | 16.8 | 34.3 | 18.5 | 41.8 | 25.3 | 39.6 | 0.3 | 10.0 | 23.4 | 36.5 | 38.6 | 49.0 | 66.0 |
| Triangle | 36.2 | 17.1 | 34.1 | 21.2 | 35.5 | 25.3 | 50.0 | 5.0 | 32.0 | 28.9 | 32.4 | 40.5 | 63.0 | 85.9 |
| StreamingMix | 39.9 | 18.5 | 34.4 | 19.8 | 42.3 | 25.9 | 47.4 | **5.3** | 85.0 | 30.0 | 38.3 | 40.1 | 48.3 | 83.3 |
| DuoAttention | 42.6 | 19.2 | **34.6** | 25.1 | 44.4 | **26.1** | **51.8** | 2.3 | **87.7** | 29.6 | 41.1 | 40.4 | 65.7 | 85.9 |
| TriangleMix | **43.4** | **20.7** | 34.5 | **25.7** | **47.3** | **26.1** | 51.1 | 4.7 | 85.3 | **30.7** | **43.6** | **40.9** | **66.7** | **87.0** |
| *Qwen2.5-7B-Instruct* | 46.8 | 24.5 | 30.7 | 32.2 | 62.2 | 22.3 | 41.3 | 13.1 | 99.3 | 24.4 | 60.7 | 42.5 | 67.0 | 88.7 |
| Streaming | 28.3 | 10.1 | 31.3 | 9.5 | 61.7 | 21.5 | 20.7 | 6.5 | 8.2 | 11.4 | 44.8 | 40.2 | 51.7 | 49.9 |
| Triangle | 36.9 | 17.4 | 27.5 | 24.0 | 48.5 | 17.9 | 38.5 | 5.6 | 43.9 | 22.3 | 39.6 | 41.2 | 65.0 | 88.1 |
| StreamingMix | 45.2 | 24.4 | **31.9** | **38.5** | 68.2 | 22.5 | **43.5** | 10.7 | **93.0** | 24.8 | **60.9** | 42.0 | 43.7 | 83.1 |
| TriangleMix | **46.1** | **24.7** | 30.6 | 32.3 | 62.6 | 22.3 | 40.9 | **13.1** | 89.3 | 23.9 | 59.6 | **42.4** | 69.3 | 88.5 |

Table 5: Comparison with static sparsity methods on Longbench.

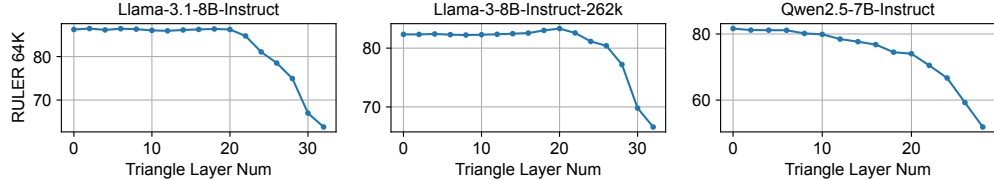

Figure 4: Average RULER score at 64K length for different $L_{\text{tri}}$ values.

## 3.3 Analysis of $L_{\text{tri}}$

In this section, we investigate the choice of $L_{\text{tri}}$ for each LLM. We evaluate all models on the 64K length tasks in the RULER benchmark, sweeping $L_{\text{tri}}$ from 0 to the total number of layers with a step size of 2. As shown in Figure 4, Llama-3.1-8B-Instruct and Llama-3-8B-Instruct-262K exhibit similar patterns: setting $L_{\text{tri}} = 20$ retains 99.7% and 100.0% of their original performance, respectively. This indicates that 62.5% of the layers can adopt the $O(N)$ Triangle Attention without noticeable degradation. In contrast, Qwen-2.5-7B-Instruct requires a less threshold: setting $L_{\text{tri}} = 8$ retains 98.2% of the performance, corresponding to a sparsity ratio of 28.6%.

| Method | *Llama-3.1-8B-Instruct* | | | *Qwen2.5-7B-Instruct* | | |
|---|---|---|---|---|---|---|
| | 32K | 64K | 128K | 32K | 64K | 128K |
| **Dense** | 44 | 179 | 750 | 40 | 156 | 646 |
| **MInference** | 94 (0.5x) | 152 (1.2x) | 229 (3.3x) | 93 (0.4x) | 151 (1.0x) | 245 (2.6x) |
| **FlexPrefill** | 32 (1.4x) | 71 (2.5x) | 220 (3.4x) | 38 (1.0x) | 99 (1.6x) | 310 (2.1x) |
| **XAttention** | 47 (0.9x) | 127 (1.4x) | 391 (1.9x) | 44 (0.9x) | 128 (1.2x) | 391 (1.7x) |
| **Triangle** | **12 (3.7x)** | **24 (7.5x)** | **49 (15.3x)** | **11 (3.7x)** | **21 (7.4x)** | **76 (8.5x)** |

Table 6: Attention kernel latency (ms) for one layer.

## 3.4 Efficiency of TriangleMix

We implement Triangle Attention using Triton (Tillet et al., 2019), with further implementation details provided in Appendix A.2. All experiments are conducted on a single NVIDIA A100 80GB GPU. Dense attention is implemented using FlashAttention (Kwon et al., 2023).

**Attention Kernel Latency.** We first benchmark the average attention kernel time per layer on Llama-3.1-8B-Instruct and Qwen2.5-7B-Instruct at sequence lengths of 32K, 64K, and 128K. As shown in Table 6, Triangle Attention achieves $3.7\times$ to $15.3\times$ speedup compared to dense attention,. The speedup mainly comes from Triangle Attention's linear complexity ($O(N)$), which makes it much more efficient than $O(N^2)$ dense attention. Besides, unlike dynamic sparsity methods, TriangleMix does not require estimating attention block indices. Its triangle attention kernel is also much simpler to implement, which further reduces overhead.

| Method | 32K | 48K | 64K | 80K | 96K | 112K | 128K |
|---|---|---|---|---|---|---|---|
| Dense | 4.1 | 7.3 | 11.2 | 15.9 | 21.3 | 27.5 | 34.5 |
| DuoAttention | 3.6 (-13%) | 5.8 (-20%) | 8.7 (-22%) | 11.7 (-26%) | 15.5 (-27%) | 19.1 (-31%) | 23.7 (-31%) |
| TriangleMix | 3.6 (-12%) | 5.9 (-19%) | 8.6 (-23%) | 11.7 (-26%) | 15.2 (-29%) | 19.1 (-31%) | 23.4 (-32%) |
| MInference | 5.5 (+34%) | 7.8 (+7%) | 10.1 (-10%) | 12.3 (-23%) | 13.4 (-37%) | 15.9 (-42%) | 18.0 (-48%) |
| Ours + MInference | 4.2 (+2%) | 6.0 (-18%) | 7.7 (-31%) | 9.5 (-40%) | 10.9 (-49%) | 12.7 (-54%) | 14.5 (-58%) |
| FlexPrefill | 3.6 (-12%) | 5.5 (-25%) | 7.7 (-31%) | 9.9 (-38%) | 12.3 (-42%) | 15.0 (-45%) | 17.8 (-48%) |
| Ours + FlexPrefill | 3.4 (-17%) | 5.2 (-29%) | 7.2 (-36%) | 9.2 (-42%) | 11.3 (-47%) | 13.6 (-51%) | 16.1 (-53%) |
| XAttention | 4.1 (-1%) | 6.6 (-10%) | 9.4 (-16%) | 12.4 (-22%) | 15.7 (-26%) | 19.3 (-30%) | 23.2 (-33%) |
| Ours + XAttention | 3.5 (-14%) | 5.5 (-24%) | 7.7 (-31%) | 10.0 (-37%) | 12.4 (-42%) | 15.1 (-45%) | 17.8 (-48%) |

Table 7: Time-to-first-token (TTFT) in seconds measured on Llama-3.1-8B-Instruct.

**End-to-End TTFT.** We measure the end-to-end time-to-first-token (TTFT) for sequence lengths of 32K, 48K, 64K, 96K, 112K, and 128K. Table 7 reports the results on Llama-3.1-8B-Instruct. TriangleMix yields a TTFT reduction of 12%–32%. Compared with DuoAttention, TriangleMix achieves similar efficiency but causes less performance degradation (see Section 3.2). In addition, DuoAttention partitions heads within the same layer into different sparsity patterns (Liu et al., 2025), which leads to imbalance under tensor parallelism, while TriangleMix avoids this issue. Moreover, TriangleMix can be seamlessly combined with dynamic attention for further efficiency gains. For instance, integrating MInference with TriangleMix lowers TTFT from 18.0s to 14.5s (a 19% reduction) at length 128K. TriangleMix combined with FlexPrefill achieves the lowest TTFT for 32K–80K inputs, whereas TriangleMix with MInference is optimal for 96K–128K inputs.

# 4 RELATED WORKS

**Static Sparsity Attention**. Early methods employ fixed sparse patterns, such as strided (Child et al., 2019), dilated (Ding et al., 2023), sliding window (Jiang et al., 2023), and mixed patterns (Beltagy et al., 2020), typically requiring training from scratch. DuoAttention (Xiao et al., 2024) introduces head-level static sparsity but needs additional offline training. Streaming pattern is a training-free approach (Xiao et al., 2023) but leads to degraded accuracy for long contexts (Li et al., 2024a). In contrast, the proposed TriangleMix attention pattern significantly mitigates performance loss and is nearly lossless in accuracy.

**Dynamic Sparsity Attention**. Existing methods such as MInference (Jiang et al., 2024a), FlexPrefill (Lai et al., 2025), and XAttention (Xu et al., 2025) dynamically identify attention blocks with high scores and restrict computation to these blocks. While this form of attention score sparsity focuses on pruning blocks with small attention scores, our work moves beyond by uncovering a different kind of sparsity, which we term decoding-time contribution sparsity. Specifically, we show that many blocks with non-trivial scores still contribute little to decoding, and thus can be safely skipped. Leveraging this property enables additional acceleration in the prefilling stage.

**Long-context LLM Inference**. FlashAttention (Dao et al., 2022) speeds up attention by reducing memory access through fused operations. PagedAttention (Kwon et al., 2023) improves decoding by managing KV cache allocation efficiently. KV cache optimizations techniques also include token-level eviction (SnapKV (Li et al., 2024b)) and query-aware cache selection (Quest (Tang et al., 2024)). TriangleMix is orthogonal to these approaches.

# 5 CONCLUSION

In this paper, we identify a new form of sparsity in the prefilling stage, termed decoding-time contribution sparsity, and introduce TriangleMix, a training-free static attention pattern. By selectively applying Triangle attention in certain layers, TriangleMix substantially reduces attention overhead while maintaining nearly lossless performance. On 128K inputs, it achieves up to a 15.3× speedup in attention computation, surpassing typical dynamic sparse methods. Moreover, TriangleMix can be seamlessly combined with dynamic sparsity, yielding an additional 6%–19% reduction in TTFT.

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

# A  APPENDIX

## A.1  DETAILS OF ATTENTION BLOCK PROBING

We divide the causal attention matrix into three distinct sections as illustrated in Figure 1a:

- Streaming section: includes attention sink and the sliding window;
- Last Q–K section: covers interactions between the last part of $Q$ and $K$, excluding the Streaming section;
- Middle Q–K section: consists of the remaining interactions between the middle parts of $Q$ and $K$.

We define the Streaming mask as:

$$M_{i,j}^{\mathrm{streaming}} = \begin{cases} 1, & i \geq j, j \leq si \\ 1, & i \geq j, i - j \leq sl \\ 0, & \text{otherwise} \end{cases}$$

where $si$ is the number of sink tokens and $sl$ is the sliding window size.

We define the Last Q–K mask as:

$$M_{i,j}^{\mathrm{last}} = \begin{cases} 1, & i \geq j, N - i < last, \\ & j > si, i - j > sl \\ 0, & \text{otherwise} \end{cases}$$

where $last \geq 1$ specifies the number of rows corresponding to the last section.

The Middle Q–K mask is defined as:

$$M_{i,j}^{\mathrm{middle}} = \begin{cases} 1, & i \geq j, N - i \geq last, \\ & j > si, i - j > sl \\ 0, & \text{otherwise} \end{cases}$$

In our gradient-based attention block probing experiments, we set the parameters to $si = 64$, $sl = 128$, and $last = 128$. We adopt a key-value retrieval task, where the language model is prompted to output the correct value starting from its very first generated token. Each input sequence contains approximately 2,000 tokens, and gradients are computed over 100 randomly generated samples. The prompt used in this task is as follows:

```
Prompt for Attention Importance Probing:

Extract the value corresponding to the specified key in the data below.

Data:
<key 1>: <value 1>
<key 2>: <value 2>
......
<key n>: <value n>

Extract the value corresponding to this key:
key: {key i}

Please directly output the corresponding value without outputing anything else.
value:
```

We randomly generate pairs of keys and values using UUID strings.

## A.2 Implementation of Triangle Attention

We implement Triangle Attention using Triton (Tillet et al., 2019), with details provided in Algorithm 1. For each row index, we apply a different attention mechanism: if the row index is less than $N - N_{\text{last}}$, we apply Streaming Attention; otherwise, we apply chunked FlashAttention to increase GPU utilization. The outputs from both segments are then merged to produce the final result.

---

**Algorithm 1** Fused Triangle Attention

---

**Input data:** $Q, K, V \in \mathbb{R}^{N \times d_h}$
**Input triangle shape:** sink token number $N_{\text{sink}}$, sliding window size $N_{\text{window}}$, last rows number $N_{\text{last}}$
**Input kernel block shape:** $B_M, B_N$
Calculate best number of splits $S$ for last row
Initialize $O \leftarrow (0)^{N \times d_h}$, output buffer $O_{\text{last}} \leftarrow (0)^{SN_{\text{last}} \times d_h}$, LSE buffer $l_{\text{last}} \leftarrow (-\inf)^{SN_{\text{last}}}$

*Parallelized in GPU*
**for** $i \leftarrow 1$ to $\lceil \frac{N - N_{\text{last}}}{B_M} \rceil + S \lceil \frac{N_{\text{last}}}{B_M} \rceil$ **do**
    Initialize $O_{\text{chip}} \leftarrow (0)^{B_M \times d_h}$, $m \leftarrow (-\inf)^{B_N}$, $s \leftarrow (0)^{B_M}$
    **if** $i \leq \lceil \frac{N - N_{\text{last}}}{B_M} \rceil$ **then**

        *Upper part: the same as streaming attention*
        Load $Q_{\text{chip}} \leftarrow Q^{iB_M:(i+1)B_M}$

        *Loop through sink tokens*
        **for** $j \leftarrow 1$ to $\lceil \frac{N_{\text{sink}}}{B_N} \rceil$ **do**
            Load $K_{\text{chip}} \leftarrow K^{jB_N:(j+1)B_N}$, $V_{\text{chip}} \leftarrow V^{jB_N:(j+1)B_N}$
            flash_attn($Q_{\text{chip}}, K_{\text{chip}}, V_{\text{chip}}, O_{\text{chip}}, m, s$)
        **end for**

        *Loop through sliding window*
        **for** $j \leftarrow \lceil \frac{N - N_{\text{window}}}{B_N} \rceil$ to $\lceil \frac{N}{B_N} \rceil$ **do**
            Load $K_{\text{chip}} \leftarrow K^{jB_N:(j+1)B_N} \in \mathbb{R}^{B \times d_h}$, $V_{\text{chip}} \leftarrow V^{jB_N:(j+1)B_N} \in \mathbb{R}^{B \times d_h}$
            flash_attn($Q_{\text{chip}}, K_{\text{chip}}, V_{\text{chip}}, O_{\text{chip}}, m, s$)
        **end for**

        *Write outputs*
        Save $O^{iB_M:(i+1)B_M} \leftarrow O_{\text{chip}}$
    **else**

        *Last rows: split in to S chunks*
        Chunk index $c \leftarrow \lfloor (i - \lceil \frac{N - N_{\text{last}}}{B_M} \rceil) / \lceil \frac{N_{\text{last}}}{B_M} \rceil \rfloor$
        Chunk offset $b \leftarrow (i - \lceil \frac{N - N_{\text{last}}}{B_M} \rceil) \mod \lceil \frac{N_{\text{last}}}{B_M} \rceil$
        Q index $i_Q \leftarrow \lceil \frac{N - N_{\text{last}}}{B_M} \rceil + b$
        Load $Q_{\text{chip}} \leftarrow Q^{i_Q B_M:(i_Q+1)B_M}$
        **for** $j \leftarrow c \lceil \frac{N}{SB_N} \rceil$ to $(c+1) \lceil \frac{N}{SB_N} \rceil$ **do**
            Load $K_{\text{chip}} \leftarrow K^{jB_N:(j+1)B_N}$, $V_{\text{chip}} \leftarrow V^{jB_N:(j+1)B_N}$
            flash_attn($Q_{\text{chip}}, K_{\text{chip}}, V_{\text{chip}}, O_{\text{chip}}, m, s$)
        **end for**

        *Write outputs*
        Save $O_{\text{last}}^{(i-i_Q)B_M:(i-i_Q+1)B_M} \leftarrow O_{\text{chip}}$
        Save $l_{\text{last}}^{(i-i_Q)B_M:(i-i_Q+1)B_M} \leftarrow \ln s + m$
    **end**
**end for**

*Merge last row output buffer*
$O^{(N - N_{\text{last}}):N} \leftarrow \text{merge\_output}(O_{\text{last}}, l_{\text{last}})$

---

| Methods | 8K | | | | 16K | | | | 32K | | | | Average |
|---|---|---|---|---|---|---|---|---|---|---|---|---|---|
| | op=2 | op=4 | op=6 | op=8 | op=2 | op=4 | op=6 | op=8 | op=2 | op=4 | op=6 | op=8 | |
| *Llama-3.1* | 0.142 | 0.050 | 0.042 | 0.040 | 0.136 | 0.058 | 0.030 | 0.030 | 0.210 | 0.080 | 0.058 | 0.036 | 0.076 |
| TriangleMix | 0.156 | 0.048 | 0.044 | 0.030 | 0.108 | 0.066 | 0.028 | 0.028 | 0.208 | 0.070 | 0.068 | 0.036 | 0.074 |
| *Qwen2.5* | 0.404 | 0.182 | 0.138 | 0.114 | 0.266 | 0.110 | 0.142 | 0.064 | 0.238 | 0.106 | 0.100 | 0.066 | 0.161 |
| TriangleMix | 0.428 | 0.196 | 0.176 | 0.114 | 0.286 | 0.134 | 0.108 | 0.068 | 0.258 | 0.110 | 0.092 | 0.056 | 0.169 |
| *DeepSeek-Distill* | 0.200 | 0.078 | 0.048 | 0.040 | 0.288 | 0.132 | 0.086 | 0.082 | 0.070 | 0.016 | 0.018 | 0.018 | 0.090 |
| TriangleMix | 0.210 | 0.088 | 0.058 | 0.032 | 0.300 | 0.110 | 0.082 | 0.088 | 0.036 | 0.022 | 0.026 | 0.022 | 0.090 |

Table 8: The evaluation results on GSM-infinite hard subset with various context lengths and operation numbers.

### A.3 REASONING BENCHMARKS

We further evaluate our method on the GSM-Infinite reasoning benchmark (Zhou et al.), which simultaneously challenges both long-context and reasoning ability of the model. GSM-Infinite constructs a complex computational graph containing both task-relevant operations and distractor operations; by varying the number of operations, we can precisely control the context length and complexity of the underlying graph. We use the hard subset of GSM-Infinite and evaluate models under context lengths of 8K, 16K, and 32K, and operation counts of 2, 4, 6, and 8. For each context–operation pair, we sample 500 problems, resulting in a total of 6,000 test instances. The evaluating metric is accuracy.

We compare full attention with our proposed TriangleMix on Llama-3.1-8B-Instruct, Qwen2.5-7B-Instruct, and a reasoning model DeepSeek-R1-Distill-Qwen-7B (Guo et al., 2025). As shown in Table 8, TriangleMix yields only $-0.002$ to $+0.008$ absolute accuracy change to dense attention, demonstrating that it preserves both long-context capacity and reasoning performance across all settings. We also observe that DeepSeek-R1-Distill-Qwen-7B underperforms its origin model Qwen2.5-7B-Instruct, possibly because the distillation process weakened some long-context abilities. A deeper investigation is left for future work.

### A.4 ACKNOWLEDGEMENT OF LLM USAGE

We used large language models to polish the writing of this paper, and all generated content was carefully reviewed to ensure precise expression.

