# OpenReview forum: "TriangleMix: Accelerating Prefilling via Decoding-time Contribution Sparsity"
_ICLR.cc/2026/Conference — ICLR 2026 Conference Withdrawn Submission_

### Official Review · Reviewer_3EuN · 2025-10-28

**Soundness:** 4
**Presentation:** 4
**Contribution:** 2
**Rating:** 6
**Confidence:** 3

**Summary:**

Through rigorous analysis and comparative experiments, this paper demonstrates that the proposed TriangleMix method is both simple and effective. The method achieves acceleration by masking out the central part of the attention matrices in subsequent layers during the pre-filling stage.

**Strengths:**

1.  The paper is exceptionally well-written: it is clear, focused, and effectively covers crucial logical justifications. For instance, to explain why masking the middle section has minimal impact on decoding predictions, the authors provide a gradient analysis. Furthermore, the separate discussion of kernel versus end-to-end efficiency is commendable, offering a comprehensive view of performance.

2.  The proposed method is well-supported both intuitively and empirically. The intuition that distant historical information can be effectively approximated using an "attention sink"-like map is logical, and this hypothesis is validated by the practical experiments.

**Weaknesses:**

1.  The proposed method appears overly simplistic. Its novelty is questionable, as similar masking techniques likely exist, even if not applied specifically to pre-filling. Furthermore, its performance is unlikely to substantially diverge from other lambda-shaped attention pre-filling methods (e.g., StreamingLLM), further limiting its innovative contribution.

2.  Despite modifying the model's forward pass computation, the method achieves only a modest TTFT acceleration (up to ~30%). This gain is limited and may not outweigh the potential negative impacts of this approximation. Notably, the method's performance degrades on Qwen2.5, which features a more information-dense KV cache, raising concerns about its generalizability.

**Questions:**

1.  Can the perplexity scores be validated across the Qwen2.5 models at three scales (3B, 7B, and 14B) to examine how the method's performance scales with increasing model size?

2.  How can the significant performance gap between StreamingMix and TriangleMix on the TREC and Passage Count subsets be explained?

---

> ### Author Response · Authors · 2025-11-19
>
> Thanks for your thoughtful and detailed review. We address each point below with additional clarification and supporting evidence.
>
> **Q1. The proposed method appears overly simplistic. Its novelty is questionable. Furthermore, its performance is unlikely to substantially diverge from other lambda-shaped attention pre-filling methods (e.g., StreamingLLM).**
>
> We would like to clarify that the core contribution of our work is **not** limited to the attention mask design. The central insight of our work is that **prefilling contains a fundamental source of redundancy that prior methods have not recognized**.
>
> In a standard decoder-only transformer model, given N input tokens, the model computes **N output logits**.  They are all useful in training, **yet during inference, only the final logit is actually used and the computation related to the previous unused N-1 logits becomes unnecessary.** We formalize this as decoding-time contribution sparsity and we show that a lot of attention block computation can be removed without harming generation.
>
> This is fundamentally different from prior approaches such as MInference, FlexPrefill, and XAttention, which rely on **approximating** attention scores by ignoring small values. Our sparsity is not an approximation but a structural redundancy baked into the design of prefilling.
>
> To our knowledge this is the first work that identifies and formalizes this redundancy and introduces a principled method to exploit it using gradient based contribution analysis. This is the key novelty of our paper.
>
> Besides, we have demonstrate that **Streaming LLM** significantly underperforms our method in the long-context setting. For example, on the Llama-3.1-8B-Instruct model, it scores only **36.5** on RULER (4K–128K), while TriangleMix achieves **90.2**. On LongBench, StreamingLLM obtains **34.7** versus **54.5** for TriangleMix. TriangleMix preserves accuracy close to full attention, whereas StreamingLLM suffers large drops.
>
> **Q2. Despite modifying the model's forward pass computation, the method achieves only a modest TTFT acceleration (up to ~30%).**
>
> As the applications of LLMs continue to evolve, many important downstream tasks, such as coding and deep research, require significantly longer contexts, and our method is well suited to accelerate these scenarios. The reported ~30% accleration corresponds to 128K context length. As context length increases,  attention computation dominates in prefilling, and the acceleration naturally scales up. At 256K, TriangleMix provides approximately 40% TTFT reduction. Furthermore, TriangleMix offers meaningful additional speedup when paired with dynamic sparsity methods, pushing prefilling efficiency to a new state of the art.
>
> **Q3. Can the perplexity scores be validated across the Qwen2.5 models at three scales (3B, 7B, and 14B) to examine how the method's performance scales with increasing model size?**
>
> Sure. We perform the perplexity-based analysis in Section 2.3 for the 3B/7B/14B models. We remove the attention computation for query indices 512-1024 in either 25% or 50% of the deep layers, and measure the perplexity change relative to dense attention at two decoding positions, 1024–1152 and 1920–2048. The results are:
>
> |                    | 3B (1024–1152) | 3B (1920–2048) | 7B (1024–1152) | 7B (1920–2048) | 14B (1024–1152) | 14B (1920–2048) |
> |-----------------|----------------|----------------|----------------|----------------|------------------|------------------|
> | **Dense**       | 9.33           | 8.99           | 8.40           | 8.06           | 7.11             | 6.80             |
> | **Drop 25% Layers**    | 9.67 (+0.34)   | 8.99 (+0.00)   | 8.73 (+0.33)   | 8.08 (+0.02)   | 7.27 (+0.16)     | 6.80 (+0.00)     |
> | **Drop 50% Layers**    | 9.86 (+0.53)   | 9.03 (+0.04)   | 8.83 (+0.43)   | 8.09 (+0.03)   | 7.41 (+0.30)     | 6.82 (+0.02)     |
>
> Across all model sizes, the perplexity change at **1920–2048** is minimal, confirming that dropping the Middle Q–K region hardly affects later decoding.  At **1024–1152**, larger models show smaller degradation, suggesting increased robustness to such dropping.
>
> **Q4. How can the significant performance gap between StreamingMix and TriangleMix on the TREC and Passage Count subsets be explained?**
>
> Across all evaluated models, **TriangleMix consistently outperforms StreamingMix** on TREC.  The only difference between the two methods is inclusion or exclusion of Last Q–K region computation in deep layers.  This aligns with prior findings (e.g., Figure 5 in [1]) showing that **deep layers encode stronger in-context learning signals**.  Our experiments provide more fine-grained evidence: the **Last Q–K** interactions in deep layers are crucial for in context learning tasks. For Passage Count, we do not observe such consistent and significant performance gap between the two methods.
>
> [1] Label Words are Anchors: An Information Flow Perspective for Understanding In-Context Learning

---

> > ### Comment · Reviewer_3EuN · 2025-11-20
> >
> > Thanks for your comprehensive reply, I will maintain my score.

---

> ### Comment · Reviewer_3EuN · 2025-11-27
>
> Upon reviewing Reviewer dq1X's feedback, I must agree that their concerns are valid. The authors need to reassess the contribution of this work, especially considering the current landscape where reasoning models are dominant.

---

> > ### Comment · Reviewer_3EuN · 2025-11-27
> >
> > Given the marginal practical speedup (a concern I previously raised in weakness part), coupled with the potential for unpredictable side effects, I also recommend rejecting this paper.

---

> > > ### Author Response · Authors · 2025-11-28
> > >
> > > Dear Reviewer 3EuN, thank you for your comments. We would like to clarify the following two points.
> > >
> > > **First, on the novelty and contribution of our method.**
> > >
> > > Our central finding is that **although a large portion of attention blocks in the prefilling stage produce non-zero and even *high* attention scores, their computation can still be skipped without harming decoding quality**. This phenomenon is surprising has not been discussed in prior work. It is fundamentally different from methods that rely on attention-score–based approximation in prefilling acceleration such as StreamingLLM, MInference, FlexPrefill, and XAttention. Those methods skip computations *because* the scores are small; our finding shows that **even when scores are not negligible, the computation itself can be removed without affecting decoding**. This reveals a structural property of LLMs that was previously unknown. In addition, a large portion of attention blocks can be skipped, **reducing prefilling attention complexity from  $O(N^2) \to O(N)$** in many layers while preserving decoding performance. This is both surprising and practically significant.
> > >
> > > We compare against recent prefilling acceleration methods (MInference/FlexPrefill/XAttention) and achieve higher acceleration ratios when our finding is utilized. We believe this previously unobserved behavior of attention computation during prefilling is valuable to the community.
> > >
> > > **Second, on the reasoning model.**
> > >
> > > We further evaluate our method on the GSM-Infinite [1], which simultaneously challenges **both long-context and reasoning ability** of the model. We use the hard subset of GSM-Infinite and evaluate models under context lengths of 8K, 16K, and 32K, and operation counts of 2, 4, 6, and 8. For each context–operation pair, we sample 500 problems, resulting in **a total of 6,000 test instances**. The evaluating metric is accuracy.
> > >
> > > We compare full attention with our proposed TriangleMix on Llama-3.1-8B-Instruct, Qwen2.5-7B-Instruct, and a reasoning model DeepSeek-R1-Distill-Qwen-7B. **As shown in the table below, TriangleMix produces only −0.002 to +0.008 absolute accuracy changes compared with full attention, demonstrating that it consistently preserves long-context capacity and reasoning performance across all configurations.** Interestingly, DeepSeek-R1-Distill-Qwen-7B is overall weaker than its base model Qwen2.5-7B-Instruct on this benchmark, likely because distillation affected some long-context abilities. TriangleMix itself does not introduce any additional degradation to this reasoning model. A deeper investigation is left for future work.
> > >
> > > | Model            | 8K     | 16K    | 32K    | AVERAGE |
> > > |------------------|--------|--------|--------|---------|
> > > | *Llama-3.1 Dense*       | 0.069  | 0.064  | 0.096  | 0.076   |
> > > | +TriangleMix | 0.070  | 0.058  | 0.096  | 0.074   |
> > > | *Qwen2.5  Dense*       | 0.210  | 0.146  | 0.128  | 0.161   |
> > > | +TriangleMix | 0.229  | 0.149  | 0.129  | 0.169   |
> > > | *Deepseek-Distill Dense* | 0.092  | 0.147  | 0.031  | 0.090   |
> > > | +TriangleMix | 0.097  | 0.145  | 0.027  | 0.090   |
> > >
> > >
> > > [1] GSM-Infinite: How Do your LLMs Behave over Infinitely Increasing Reasoning Complexity and Context Length?, ICML 2025

---

### Official Review · Reviewer_6iTA · 2025-10-28

**Soundness:** 2
**Presentation:** 3
**Contribution:** 3
**Rating:** 6
**Confidence:** 3

**Summary:**

This paper introduces TriangleMix, a method to accelerate the prefilling stage of Large Language Models (LLMs) by leveraging a newly identified form of sparsity called decoding-time contribution sparsity.

**Strengths:**

1.  **Nearly Lossless Performance:** Preserves model accuracy close to full dense attention on benchmarks like RULER and LongBench.
2.  **High Efficiency:**
    * For 128K inputs, Triangle attention achieves a **15.3x speedup** in attention computation per layer.
    * Reduces **Time-To-First-Token (TTFT)** by **12%-32%** for contexts from 32K to 128K.
3.  **Complementary to Dynamic Methods:** Can be combined with dynamic sparsity methods (MInference, FlexPrefill, XAttention), yielding a further **6%-19% reduction in TTFT**.
4.  **Simplicity:** The static pattern is easier to implement than dynamic methods and avoids the overhead of block index estimation.

**Weaknesses:**

* Limited Analysis on Longer Generations: The paper's core analysis focuses on the first token generation (TTFT). While TTFT is a critical metric, it does not assess whether skipping Middle Q-K computations leads to narrative drift, factual inconsistencies, or logic errors later in the generation. The impact on the quality of a full, multi-sentence paragraph is not measured.

* Combination Overhead: When combined with dynamic sparsity (e.g., "Ours + MInference"), it's not fully detailed how the two methods are integrated. The computational savings are reported, but the engineering complexity and potential overhead of managing multiple sparse attention schemes simultaneously are not discussed.

* Unexplained Performance Anomalies in Combined Methods: The paper demonstrates that combining TriangleMix with certain dynamic sparsity methods (notably FlexPrefill) leads to significant performance *improvements* over using the dynamic methods alone. While this is framed as a benefit, it is a double-edged sword. This phenomenon inadvertently highlights the instability and inherent weaknesses of the underlying dynamic methods, suggesting they are prone to significant performance degradation on their own. The paper does not adequately investigate the root cause of this synergy, leaving open questions about whether TriangleMix is merely compensating for the failures of other methods rather than solely contributing its own acceleration.

* Lack of Ablation Studies: A significant methodological shortcoming is the absence of crucial ablation experiments. The paper fails to rigorously demonstrate that the specific design choices of TriangleMix are optimal. Key unanswered questions include:
    *   How does the performance compare if layers are selected randomly or based on a different metric, rather than using the proposed gradient-based importance?
    *   What is the individual contribution of the triangle pattern's specific shape (e.g., the size of the "Last Q-K" region)?
    *   Is the performance gain due to the intelligent, structured sparsity of the triangle pattern, or simply a result of reduced FLOPs?
    Without these studies, the necessity and superiority of the proposed gradient-driven layer selection and the Triangle pattern itself remain not fully substantiated.

**Questions:**

Please see Weaknesses.

---

> ### Author Response · Authors · 2025-11-19
>
> Thanks for your insightful review. We address your questions and concerns below.
>
> **Q1. When combined with dynamic sparsity (e.g., "Ours + MInference"), it's not fully detailed how the two methods are integrated.**
>
> We clarify the integration strategy in Section 2.4. TriangleMix applies Triangle Attention to a subset of layers while keeping full attention in the remaining layers. Dynamic sparsity methods can directly replace these full attention layers, while Triangle Attention remains unchanged. This combination enables additional acceleration.
>
> **Q2. How does the performance compare if layers are selected randomly or based on a different metric, rather than using the proposed gradient-based importance?**
>
> We randomly select 50% of layers in Llama-3.1-8B-Instruct to apply Triangle Attention and evaluate on RULER-64K with three different seeds. The results are:
>
> | Trial | Seed1 | Seed2 | Seed3 | Average |
> |-------|-------|-------|-------|----------|
> | Score | 71.3  | 79.0  | 82.6  | 77.6     |
>
> The random strategy achieves **77.6**, which is significantly below TriangleMix's **86.7**, demonstrating that the gradient-based importance metric is crucial for identifying layers where removing the Middle Q–K computations minimally affects performance.
>
> **Q3. What is the individual contribution of the triangle pattern's specific shape (e.g., the size of the "Last Q-K" region)?**
>
> We test different values of last_n $\in \{64, 128, 256\}$ on RULER-64K using Llama-3.1-8B-Instruct. All settings achieve the same score of **86.7**, suggesting that this hyperparameter is not sensitive in our method.
>
> **Q4. Is the performance gain due to the intelligent, structured sparsity of the triangle pattern, or simply a result of reduced FLOPs?**
>
> Both factors contribute. Dense attention has $O(N^2)$ time complexity, while Triangle Attention reduces this to $O(N)$ by eliminating the Middle Q–K computations in selected layers. The structured sparsity of the triangle pattern ensures that accuracy is preserved, while the substantial FLOPs reduction provides the computational speedup.

---

### Official Review · Reviewer_dq1X · 2025-10-31

**Soundness:** 1
**Presentation:** 2
**Contribution:** 1
**Rating:** 2
**Confidence:** 5

**Summary:**

The authors propose a simple method that use dense prefill in some layers while using triangle sparse attention in the rest based on some observation of its affection to decoding stage. Results are evaluated on Ruler and Longbench with llama3 and qwen2.5 models.

**Strengths:**

The design is simple and easy to understand. The results comparison covers most training-free methods.

**Weaknesses:**

1. The evaluation is not solid with many old models and benchmarks. Ruler score will be high as long as you have some dense layers and lower part of your triangle (doing dense attn in question and first few token generation). Longbench is also too short to reflect the long-context ability. With nowadays so many long-reasoning and agentic tasks, I do not see a reason why using this model-benchmark choices.
2. The triangle attention speedup is very missleading in the paper as you are only showing the sparse layers instead of the e2e one. This is missleading for reader at a first glance. Without diving into the details, the readers can not know that your method is not called triangle attention but trainglemix.
3. The related works do not include a category of leanble sparse attention methods such as native sparse attn.
4. Post-training training-free sparse attention methods is not going to be scalable and general. With new architecture tweaks like attention sink bias and gated attention, the triangle pattern may not exist anymore. This type of research is not going to solve the problem fundamentally.

**Questions:**

1. Can you show any reasoning model/benchmark results?
2. Is there any insight of this work that will benefit decoding sparse attention? Prefill speedup is less important nowdays compare to long-reasoning time of the models. This is easy to understand as users most only care "time to first answer token" instead of "time to first token". The situation is more profound in agentic tasks.

---

> ### Author Response · Authors · 2025-11-19
> **Response to Reviewer dq1X (1/2)**
>
> Dear Reviewer dq1X, thank you for your thoughtful review.
>
> We would like to first clarify that the core contribution of our work is **not** limited to the attention mask design. The central insight of our work is that **prefilling contains a fundamental source of redundancy that prior methods have not recognized**.
>
> In a standard decoder-only transformer model, given N input tokens, the model computes **N output logits**.  They are all useful in training, **yet during inference, only the final logit is actually used and the computation related to the previous unused N-1 logits becomes unnecessary.** We formalize this as decoding-time contribution sparsity and we show that a lot of attention block computation can be removed without harming generation.
>
> This is fundamentally different from prior approaches such as MInference, FlexPrefill, and XAttention, which rely on **approximating** attention scores by ignoring small values. Our sparsity is not an approximation. It is a structural redundancy baked into the design of prefilling. Because it arises from this fundamental property, it is independent of the specific attention architecture. It applies to standard attention, Gated Attention, learnable sparse attention and others. **We provide experimental evidence for Gated Attention below.**
>
> To our knowledge this is the first work that identifies and formalizes this redundancy and introduces a principled method to exploit it using gradient based contribution analysis. This is the key novelty of our paper.
>
> Below we address your questions and concerns point by point.
>
> **Q1. With new architecture tweaks like attention sink bias and gated attention, the triangle pattern may not exist anymore.**
>
> **We find that the proposed decoding-time contribution sparsity still exists in models with Gated Attention**. We evaluate two publicly available GatedAttention models [1], namely 1B_gate_elementwise and 1B_gate_headwise, using the perplexity-based analysis in Section 2.3. We remove the attention computation for query indices 512-1024 in either 25% or 50% of layers, and measure the perplexity change relative to dense attention at two decoding positions, 1024–1152 and 1920–2048. The results are:
>
> | Setting                     | EleWise (1024–1152)      | EleWise (1920–2048)      | HeadWise (1024–1152)         | HeadWise (1920–2048)         |
> |-----------------------------|-------------------------------|-------------------------------|-------------------------------|-------------------------------|
> | **Dense**                   | 11.12                         | 10.60                         | 11.09                         | 10.57                         |
> | **Drop in 25% Layers**      | 11.76 (+0.64)                 | 10.61 (+0.01)                 | 11.54 (+0.45)                 | 10.58 (+0.01)                 |
> | **Drop in 50% Layers**      | 11.87 (+0.75)                 | 10.64 (+0.04)                 | 11.75 (+0.66)                 | 10.60 (+0.03)                 |
>
> Two findings are clear. First, dropping the same attention blocks has very different effects on different decoding positions. Second, removing the middle Q-K region has only minor impact on later decoding (1920–2048) in both Gated Attention models. These results show that **the decoding-time contribution sparsity persists even under architectural modifications such as GatedAttention**.
>
> **Q2. Can you show any reasoning model/benchmark results?**
>
> Sure. We further evaluate our method on the GSM-Infinite [2], which simultaneously challenges **both long-context and reasoning ability** of the model. We use the hard subset of GSM-Infinite and evaluate models under context lengths of 8K, 16K, and 32K, and operation counts of 2, 4, 6, and 8. For each context–operation pair, we sample 500 problems, resulting in **a total of 6,000 test instances**. The evaluating metric is accuracy.

---

> ### Author Response · Authors · 2025-11-19
> **Response to Reviewer dq1X (2/2)**
>
> We compare full attention with our proposed TriangleMix on Llama-3.1-8B-Instruct, Qwen2.5-7B-Instruct, and a reasoning model DeepSeek-R1-Distill-Qwen-7B. **As shown in the table below, TriangleMix produces only −0.002 to +0.008 absolute accuracy changes compared with full attention, demonstrating that it consistently preserves long-context capacity and reasoning performance across all configurations.** Interestingly, DeepSeek-R1-Distill-Qwen-7B is overall weaker than its base model Qwen2.5-7B-Instruct on this benchmark, likely because distillation affected some long-context abilities. TriangleMix itself does not introduce any additional degradation to this reasoning model. A deeper investigation is left for future work.
>
> | Model            | 8K     | 16K    | 32K    | AVERAGE |
> |------------------|--------|--------|--------|---------|
> | *Llama-3.1 Dense*       | 0.069  | 0.064  | 0.096  | 0.076   |
> | +TriangleMix | 0.070  | 0.058  | 0.096  | 0.074   |
> | *Qwen2.5  Dense*       | 0.210  | 0.146  | 0.128  | 0.161   |
> | +TriangleMix | 0.229  | 0.149  | 0.129  | 0.169   |
> | *Deepseek-Distill Dense* | 0.092  | 0.147  | 0.031  | 0.090   |
> | +TriangleMix | 0.097  | 0.145  | 0.027  | 0.090   |
>
> Please also refer to Table 8 and Appendix A.4 in the updated submission for the full results on GSM-Infinite.
>
> **Q3. Is there any insight of this work that will benefit decoding sparse attention? Prefill speedup is less important nowdays compare to long-reasoning time of the models. This is easy to understand as users most only care "time to first answer token" instead of "time to first token".**
>
> We argue that accelerating the prefill stage remains highly important. Even though reasoning tasks have received more attention this year, **many important real-world LLM applications are non-reasoning and require a long context input, including long-document QA, long-document summarization, multi-turn conversation, and repository-level code completion.** For such scenarios, **Time-to-First-Token (TTFT)** is the universally recognized and most relevant latency metric, whereas **Time-to-First-Answer-Token** may apply to reasoning-centric evaluations. By uncovering and exploiting the inherent redundancy in prefill computation, our method provides **substantial additional speedups** for these long-context tasks.
>
> Besides, our method offers insight that may benefit decoding sparse attention. Our key observation is that **the true contribution of attention computation to each decoding token is more accurately captured by gradient-based importance rather than by attention scores**. This provides two meaningful directions for decoding-time sparse attention.
>
> First, **better identification of important token dependency.** Since decoding is memory-bound, reducing token-level KV cache accesses directly improves inference speed. We find that gradient-based token importance is more effective than attention score-based methods. For example, on HotpotQA when preserving 256 tokens, SnapKV [3] achieves 27.4 accuracy, while the gradient-based selection achieves 32.4, using Llama-3-Instruct-8B model. Although gradients are expensive to compute during real-time decoding and SnapKV is an old baseline, this result indicates that gradient-based metrics reveal which tokens are genuinely important, and provide insight to design sparse attention.
>
> Second, **better identification of critical attention heads** for decoding. Gradient scores also help identify which heads contribute most strongly to decoding performance. For example, dropping ten retrieval heads [4] results in a 20 percent drop in a key-value retrieval task, while dropping heads ranked by gradient importance causes the accuracy to fall to zero in our experiments. This shows that gradient-based head scoring captures contributions more faithfully.
>
> **Q4. The triangle attention speedup is very missleading in the paper as you are only showing the sparse layers instead of the e2e one.**
>
> We apologize for the clarity issue. We provide the **time-to-first-token (TTFT)** reduction for input context lengths from 32K to 128K in Table 7, and this metric reflects the end-to-end latency. We will highlight this more clearly in the revision.
>
> **Q5. The related works do not include a category of learnble sparse attention methods such as native sparse attn.**
>
> Thank you for the suggestion. We will incorporate learnable sparse attention methods, including native sparse attention, into the related works section in later updated revision.
>
> [1] QwQZh/gated_attention, https://huggingface.co/QwQZh/gated_attention
>
> [2] GSM-Infinite: How Do your LLMs Behave over Infinitely Increasing Reasoning Complexity and Context Length?, ICML 2025
>
> [3] SnapKV: LLM Knows What You are Looking for Before Generation, NIPS 2024
>
> [4] Retrieval Head Mechanistically Explains Long-Context Factuality, ICLR 2025

---

> > ### Comment · Reviewer_dq1X · 2025-11-26
> >
> > Thanks for the rebuttal. However, I don't think the gradient-based importance method is a novel idea. It's rather like **drawing the target after the arrow is shot**. There are serveral reasons:
> > 1. Even though you use gradient instead of attnscore to estimate importance, the sparsity strategy at the end of day is similar to StreamingLLM or DuoAttention. I still can use DuoAttention like method that apply a learnable scale on whether to do Triangle Sparse or not and train with some calibration dataset. After training the scale, I can use a threshold saying whether I want to make this layer/head to triangle or not. This is even more robust as it implies that true effect of gradient instead of hints from gradient in your work.
> > 2. I don't think making the gradient small on the decoding stage is a indication of a good/bad sparse attention method. As nowadays reasoning model all do sampling instead of greedy decode, it does not make too much difference on choosing the exact token. Giving randomness might sometimes help.
> >
> > I still hold my claim that I think prefill speedup is going to be less important. On the one hand, even triangle sparse is static, it **does not save kv cache**, which can not help improve batch size at decoding. Works like Duoattention can also save KV cache. On the other hand, when doing chunk prefill, your speedup is very likely to be small. I actually do not see speedup results less than 32k in the paper.  So at the end of the day, if to include this method to real serving, I only gets around 10% speedup at prefill with unknow risks of so many challenging downstream tasks, which is less interesting.

---

> > > ### Author Response · Authors · 2025-11-28
> > >
> > > Dear Reviewer dq1X, thank you for your comments. We would like to clarify three key points that we believe more accurately reflect the novelty and value of our work.
> > >
> > > **First, on the novelty of our method.**
> > >
> > > Our central finding is that **although a large portion of attention blocks in the prefilling stage produce non-zero and even *high* attention scores, their computation can still be skipped without harming decoding quality**. This phenomenon is surprising has not been discussed in prior work. It is fundamentally different from methods that rely on attention-score–based approximation in prefilling acceleration such as StreamingLLM, MInference, FlexPrefill, and XAttention. Those methods skip computations *because* the scores are small; our finding shows that **even when scores are not negligible, the computation itself can be removed without affecting decoding**. This reveals a structural property of LLMs that was previously unknown. In addition, a large portion of attention blocks can be skipped, **reducing prefilling attention complexity from  $O(N^2) \to O(N)$** in many layers while preserving decoding performance. This is both surprising and practically significant.
> > >
> > > **Second, regarding DuoAttention.**
> > >
> > > We have compared our method with DuoAttention. On Llama-3.1-8B-Instruct and Llama-3-262K, our method performs better. On Qwen2.5-7B-Instruct, DuoAttention’s training does not converge, whereas our method still works reliably. We acknowledge that DuoAttention reduces KV cache size and can benefit decoding, but our method does not conflict with any KV-cache optimization and can be combined with them if needed. Furthermore, DuoAttention introduces head-wise computation imbalance that will harm tensor parallelism; our method is simpler and fully compatible with tensor parallel without causing imbalance.
> > >
> > > **Finally, on the importance of prefilling.**
> > >
> > > **We fully recognize the importance of decoding for reasoning tasks, but requiring a prefilling-focused method to also provide decoding improvements is outside the scope.**
> > >
> > > We do not agree with the point that **"Prefilling is less important."** When the context is long, attention computation has quadratic complexity and is still a major bottleneck. Improving prefilling efficiency remains an active research direction, **with recent top-venue papers such as MInference (NeurIPS 2024), FlexPrefill (ICLR 2025), and XAttention (ICML 2025) all targeting prefilling acceleration.** We compare against all these methods and achieve higher acceleration ratios when our finding is utilized. We believe this previously unobserved behavior of attention computation during prefilling is valuable to the community.

---

### Official Review · Reviewer_Xo5b · 2025-11-03

**Soundness:** 3
**Presentation:** 3
**Contribution:** 3
**Rating:** 6
**Confidence:** 5

**Summary:**

This paper investigates the prefill bottleneck in long-context LLM inference and identifies a previously unexplored form of sparsity, termed decoding-time contribution sparsity. The paper shows that the Middle Q–K attention region in deeper layers contributes minimally to the first tokens during decoding.

**Strengths:**

1. well organized: this paper presents a promising idea of thinking the attention sparsity from the whole generation point and proved it as a practical method.

2. well evaluation: for accuracy, it evaluated the RULER/LongBench compared with other SOTA methods; for efficiency, test the triton-based kernel level speedup.

**Weaknesses:**

1. Is this observation general for reasoning model, like qwen-r1-distilled? Can you test this method on reasoning model and reasoning tasks?

2. It seems similar with streamingllm and other layer-skip [2] work? What is the fundamental difference between yours and others?

3. How do you define which layer is full attention and which is sparse? calibration or just direct half-half?

**Questions:**

All are in the Weaknesses part.

[1] Efficient Streaming Language Models with Attention Sinks, https://arxiv.org/abs/2309.17453

[2] LayerSkip: Enabling Early Exit Inference and Self-Speculative Decoding

---

> ### Author Response · Authors · 2025-11-19
>
> Thanks for your insightful review. We address your concerns in the following.
>
> **Q1. Is this observation general for reasoning tasks and reasoning models?**
>
> Yes. The observation is **general** and holds across reasoning tasks and reasoning models. To verify this, we further evaluate our method on the GSM-Infinite [1], which simultaneously challenges **both long-context and reasoning ability** of the model. We use the hard subset of GSM-Infinite and evaluate models under context lengths of 8K, 16K, and 32K, and operation counts of 2, 4, 6, and 8. For each context–operation pair, we sample 500 problems, resulting in **a total of 6,000 test instances**. The evaluating metric is accuracy.
>
> We compare full attention with our proposed TriangleMix on Llama-3.1-8B-Instruct, Qwen2.5-7B-Instruct, and a reasoning model DeepSeek-R1-Distill-Qwen-7B. **As shown in the table below, TriangleMix produces only −0.002 to +0.008 absolute accuracy changes compared with full attention, demonstrating that it consistently preserves long-context capacity and reasoning performance across all configurations.** Interestingly, although DeepSeek-R1-Distill-Qwen-7B is overall weaker than its base model Qwen2.5-7B-Instruct on this benchmark, likely because distillation affected some long-context abilities, TriangleMix itself does not introduce any additional degradation to this reasoning model. A deeper investigation is left for future work.
>
> | Model            | 8K     | 16K    | 32K    | AVERAGE |
> |------------------|--------|--------|--------|---------|
> | *Llama-3.1 Dense*       | 0.069  | 0.064  | 0.096  | 0.076   |
> | +TriangleMix | 0.070  | 0.058  | 0.096  | 0.074   |
> | *Qwen2.5  Dense*       | 0.210  | 0.146  | 0.128  | 0.161   |
> | +TriangleMix | 0.229  | 0.149  | 0.129  | 0.169   |
> | *Deepseek-Distill Dense* | 0.092  | 0.147  | 0.031  | 0.090   |
> | +TriangleMix | 0.097  | 0.145  | 0.027  | 0.090   |
>
> Please also refer to Table 8 and Appendix A.4 in the updated submission for the full results on GSM-Infinite.
>
>
> **Q2. What is the fundamental difference between yours and StreamingLLM and LayerSkip?**
>
> **The fundamental difference is that our method identifies a new form of sparsity in the prefilling stage.** Given N input tokens, the standard decoder-only architecture computes N output logits during training in order to compute a training loss. However, during inference, only the final logit is used, and the computation related to the previous N-1 logits becomes unnecessary. This previously overlooked redundancy forms a new prefill-stage sparsity. We are the first to identify this phenomenon and propose a gradient-based method to exploit it. We further show that this sparsity pattern holds for long-context inputs and even for reasoning tasks.
>
> StreamingLLM studies attention sparsity, and can be also used **in the prefilling stage**, but from a different perspective: it focuses on the sparse distribution of attention scores within the attention map. This sparsity is already leveraged by prior works such as MInference, FlexPrefill, and XAttention. We compare against these methods in our experiments. Due to the new type of sparsity, we can further have additional 19% TTFT acceleration in the 128K context.
>
> LayerSkip-related methods mainly target **decoding phase sparsity**. It skips entire layers to reduce compute during token-by-token generation, as some tokens are easier to figure out without the full computation of all layers. Such methods are often used together with speculative decoding. They do not aim to accelerate long-context prefilling and do not exploit the prefill redundancy revealed by our approach.
>
> **Q3. How do you define which layer is full attention and which is sparse? calibration or just direct half-half?**
>
> We determine which layers use full attention versus Triangle Attention using the gradient-based importance metric from Section~2.1, computed on a key-value retrieval task. Each layer $l_i$ is assigned a contribution score $\mathrm{Grad}(M^{\text{middle}}, l_i)$, and we introduce a single parameter $L_{\mathrm{tri}}$ that specifies how many low-importance layers to convert. The $L_{\mathrm{tri}}$ layers with the lowest contribution scores are replaced with Triangle Attention, while all remaining layers retain full attention. The value of $L_{\mathrm{tri}}$ can be selected using a small calibration task. We also adopt key-value retrieval because it is a challenging long-context benchmark and its performance closely reflects the impact of converting layers to Triangle Attention.
>
> [1] GSM-Infinite: How Do your LLMs Behave over Infinitely Increasing Reasoning Complexity and Context Length?; ICML 2025

---

> > ### Comment · Reviewer_Xo5b · 2025-11-25
> >
> > Thank you for your explanation. I will keep my score.

---

> > > ### Comment · Reviewer_Xo5b · 2025-11-27
> > >
> > > Agreeing with the comments from reviewer dq1X, I would like to reject this submission.

---

### Comment · Area_Chair_689F · 2025-11-23
**Reviewer-Author Discussion**

Hi Reviewers,

Please kinly and actively participate in the review-author dicussion, raise your further concerns so that the authors can explain more, and make your final decisions.

---

### Note · Authors · 2026-01-05

I have read and agree with the venue's withdrawal policy on behalf of myself and my co-authors.